# Enabling low-cost and robust essentiality studies with high-throughput transposon mutagenesis (HTTM)

**Antoine Champie, Amélie De Grandmaison, Simon Jeanneau, Frédéric Grenier, Pierre-Étienne Jacques, Sébastien Rodrigue** *

Département de biologie, Université de Sherbrooke, Sherbrooke, Québec, Canada

* sebastien.rodrigue@usherbrooke.ca

**Data Availability Statement:** Raw Illumina sequencing data is available on SRA at the following address: https://www.ncbi.nlm.nih.gov/sra/PRJNA896618.

## Abstract

Transposon-insertion sequencing (TIS) methods couple high density transposon mutagenesis with next-generation sequencing and are commonly used to identify essential or important genes in bacteria. However, this approach can be work-intensive and sometimes expensive depending on the selected protocol. The difficulty to process a high number of samples in parallel using standard TIS protocols often restricts the number of replicates that can be performed and limits the deployment of this technique to large-scale projects studying gene essentiality in various strains or growth conditions. Here, we report the development of a robust and inexpensive High-Throughput Transposon Mutagenesis (HTTM) protocol and validate the method using *Escherichia coli* strain BW25113, the parental strain of the KEIO collection. HTTM reliably provides high insertion densities with an average of one transposon every $\leq 20$bp along with impressive reproducibility (Spearman correlation coefficients >0.94). A detailed protocol is available at protocol.io and a graphical version is also included with this article.

## Introduction

The combination of transposon mutagenesis with next-generation sequencing revolutionized the study of gene essentiality [1–4]. This approach is based on the random integration of transposons within the genomes of a large population of cells that are next grown in selective conditions prior to transposon-insertion sequencing (TIS). Cells harboring deleterious insertions become less abundant or disappear from the population, revealing important or essential genes, while insertions persist in dispensable regions. By providing a high number of transposon insertions, TIS allows a genome-wide evaluation of the importance of virtually any genomic feature. TIS has been adapted for a wide variety of cell types and was used to investigate the underlying genetic determinants of various phenotypes such as growth in specific media [2], motility [5], pathogenicity [6], as well as cell density [7].

Many TIS protocols have been described using different strategies for transposon delivery in bacterial cell populations [8], ranging from conjugative transfer of a suicide plasmid [9] to

**Funding:** This research was enabled in part by support provided by the Centre de recherche du CHUS (https://www.crchus.ca/en/home) and by the Université de Sherbrooke (https://www.usherbrooke.ca/) awarded to PEJ and SR. PEJ and SR both hold a Fonds de Recherche du Québec – Santé (FRQS) Research Scholar Career Award (https://frq.gouv.qc.ca/sante/). The funders had and will not have a role in study design, data collection and analysis, decision to publish, or preparation of the manuscript.

**Competing interests:** The authors have declared that no competing interests exist.

the electroporation of transposomes, which consist of purified transposases in complex with transposon DNA [10]. Several transposases have also been used, each with their specific characteristics including transposition efficiency, transposon recognition site identity, and target sequence biases [11]. Nevertheless, in most cases insertion site preferences are subtle and insertions occur essentially at random positions in the genome [12]. The widely used Tn5 transposase, for instance, recognizes an optimized "mosaic" inverted repeat (5′ – CTGTCTCTTATACACATCT–3′) [13] around the transposon. Which is then inserted with a slight preference for a 19 bp sequence with a more conserved 9 bp core 5′-G(CT)(CT)(CT) (AT)(AG)(AG)(AG)C-3′ [14]. This insertion bias is so light that in most cases the Tn5 transposase is considered to insert transposons essentially at random [13]. Depending on the selected protocol and transposase, the complexity and cost of these approaches vary considerably but the number of samples that can be processed simultaneously remains usually low, mostly due to the lack of protocol optimization for high-throughput applications. Addressing this limitation would greatly facilitate large-scale or systematic gene essentiality projects.

We have thus developed a new high-throughput and inexpensive method for TIS. This "High-Throughput Transposon Mutagenesis" (HTTM) protocol is divided into three main steps: 1) mutagenesis, 2) DNA extraction, and 3) sequencing library preparation (Fig 1). Each step has been optimized to obtain on average one transposon insertion per ≤20bp using *Escherichia coli* samples. A single person can process more than 960 samples per week without any specialized equipment. The total cost of the procedure per sample of a 96-well plate, from the initial bacterial culture inoculation to completed sequencing library preparation, is below 3 $ and the hands-on time is approximately 4 minutes. HTTM not only shows an unprecedented capacity to handle numerous samples simultaneously, but also constitutes a robust, inexpensive, and time-effective alternative over conventional TIS methods [15] (S2 File).

## Mutagenesis

The HTTM mutagenesis step takes advantage of the *E. coli* MFD*pir* [16] strain as a conjugative transfer donor chassis strain containing the broad host range RP4 conjugative machinery integrated into its genome, a deletion of the *dapA* gene resulting in diaminopimelic acid auxotrophy, and a π replication protein cassette required for the proper replication of $oriV_{R6K}$ replicons. Importantly, the *E. coli* MFD*pir* strain is free of the Mu bacteriophage that could interfere with the analysis if inserted into the target strain's genome along with the transposon [16]. A transposon mutagenesis plasmid, pFG051 (Fig 2A), was introduced in the MFD*pir* strain and contains an $oriT_{RP4}$ sequence enabling its conjugative transfer into bacterial cells targeted for mutagenesis, a gene encoding a hyperactive Tn5 transposase [17] under the control of a CI-repressed promoter, as well as a transposon conferring spectinomycin resistance. Finally, the repressor plasmid pAC017_cI was also introduced in MFD*pir* along with pFG051 to express the wild-type version of the λ bacteriophage *cI* repressor (Fig 2B). In contrast to the thermosensitive *cI*857 variant commonly found in molecular biology laboratories worldwide, the wild-type CI repressor can fully repress its target promoter even at a temperature of 37°C. The resulting strain, eAC494, was used as the donor for every transposon mutagenesis experiment performed in this article. Since the expression of the transposase is tightly repressed in eAC494, early transposition events are prevented, ensuring that transposon insertions occur in the target bacterial population (Fig 2C). The target used in all experiments described in this article was an *E. coli* BW25113 derivative in which a *neo*(kanamycin resistance gene)-*sgfp* cassette was inserted into the *lacZ* gene. After conjugation, insertion mutants can be grown and sub-cultured in a selective medium to eliminate target cells with a transposon interrupting an

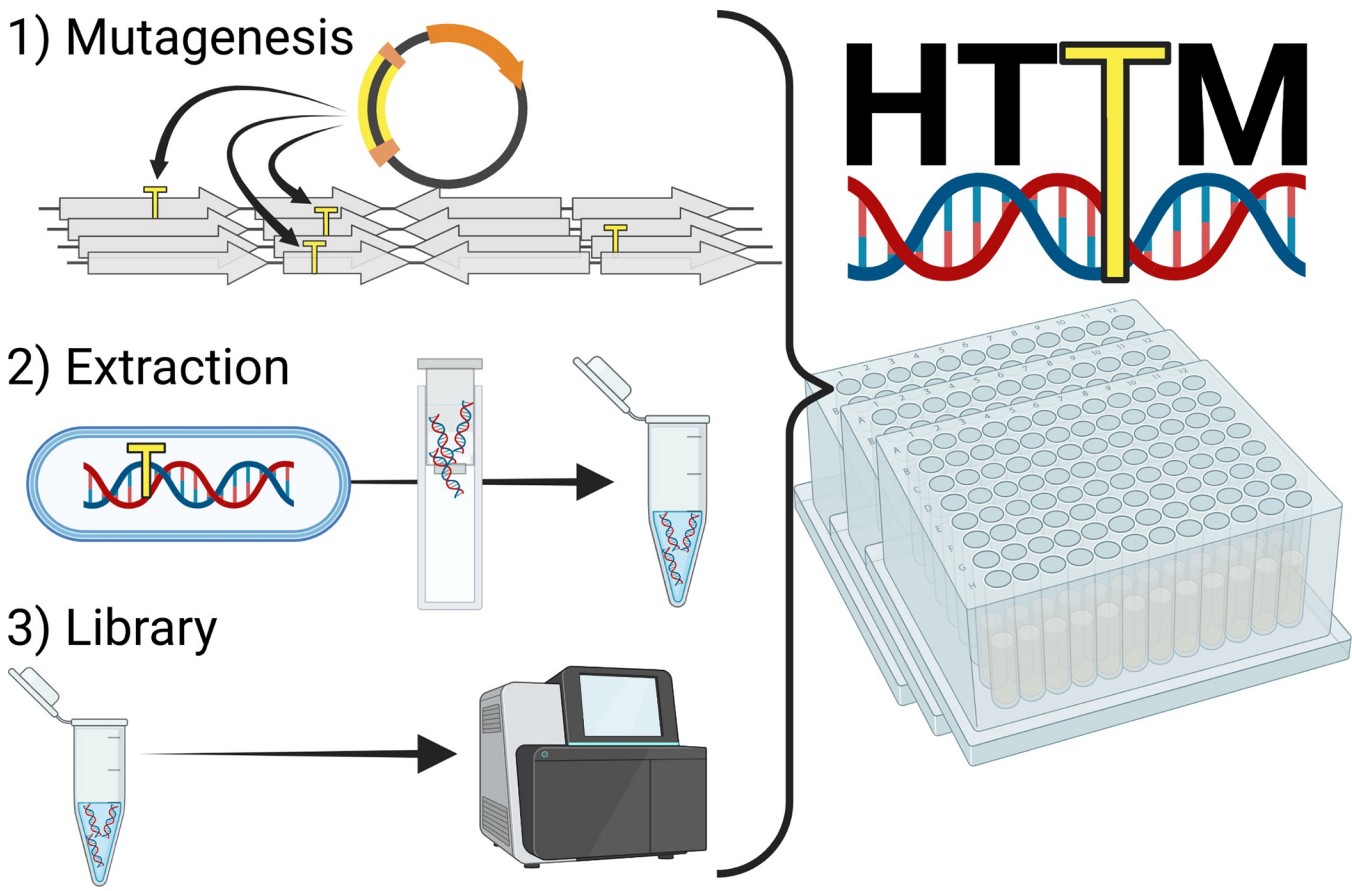

**Fig 1. Overview of the three main steps of the HTTM protocol. 1)** Transposons (yellow) are inserted at virtually random positions into the target genomes, creating a mutant pool. **2)** Genomic DNA of the mutant pool is then extracted and **3)** subjected to Illumina sequencing library preparation. All steps are conveniently performed in a 96-well plate format.

essential gene or that simply did not receive a transposon, as well as the eAC494 donor strain which is subjected to diaminopimelic acid starvation.

Bacterial conjugation could in principle be carried directly in broth using 96-well plates to easily increase the throughput of transposon mutagenesis. However, this approach has proven to be inefficient with the broad host range conjugative plasmid RP4 and would not provide enough mutants to reach high insertion densities (Fig 3A). In contrast, transfer rates are particularly high when the eAC494 and the target cells are combined on a solid support. To avoid the technical difficulties and costs associated with the deposition of conjugation mixture on cellulose filters or the inconvenient format of Petri dishes, molten agar medium was poured at the bottom of deep-well plates and allowed to solidify and dry for a few days under sterile conditions. Conjugative transfer on these "agar plugs" was then thoroughly optimized to consistently obtain ≥15 million mutants per well. Under optimized agar plug drying conditions, a conjugation mixture of 50 μl can be quickly absorbed by a dried 300 μl agar plug to generate the highest number of mutants (Fig 3B). However, this parameter may require optimization according to the exact model of deep-well plates and drying conditions. The impact of temperature on the number of insertion mutants was also evaluated, showing slightly higher yields at 37°C (Fig 3C). The donor-to-target (volume/volume) ratio was an important factor contributing to high conjugation rates. While a 10:1 ratio proved to be slightly more efficient at

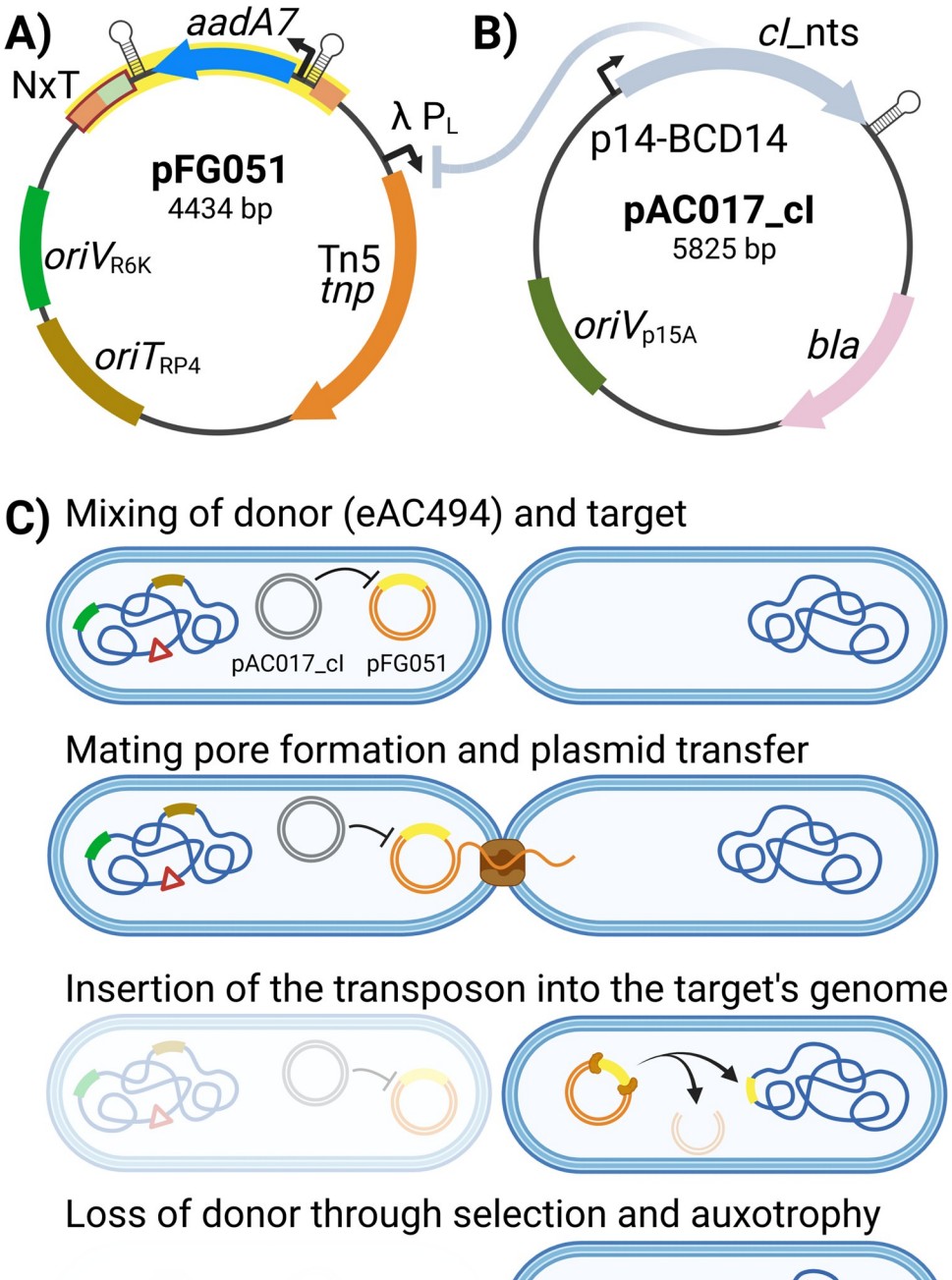

**Fig 2. HTTM relies on the conjugative delivery of a specialized transposon mutagenesis plasmid. A)** Map of the pFG051 suicide transposition plasmid. The transposon (yellow) consists of the *aadA7* gene, conferring resistance to spectinomycin driven by a constitutive promoter, flanked by two Tn*5* inverted repeats (light orange boxes) and transcription terminators (stemloops). The transposon also contains an Illumina Nextera adapter sequence (light green) adjacent to one of the two Tn*5* inverted repeats, forming the complete Nextera adapter used in library preparation (NxT, red). pFG051 depends on the conditional R6K replication origin ($oriV_{R6K}$) that is only active in the presence of the π protein exclusively expressed in the *E. coli* eAC494 donor strain. The origin of transfer ($oriT_{RP4}$) enables the mobilization of pFG051 by the broad host range RP4 conjugative machinery. The Tn*5* transposase is under the control of a wild-type CI λ repressor only present in the eAC494 donor strain. **B)** Plasmid pAC017_cI constitutively expresses the CI repressor in the eAC494 donor strain under the control of the p14-BCD14 promoter.

pAC017_cI contains the *bla* gene conferring resistance to ampicillin and a low copy replication constitutive replication origin (*oriV*p15A). cI_nts = *cI*_non thermo sensitive. **C)** Main steps involved in the conjugative delivery of the transposon mutagenesis plasmid pFG051. The donor strain (eAC494) contains genomic integration of the RP4 conjugative machinery (brown), the *pir* gene (dark green), and a *dapA* deletion (red triangle) as well as the λ *cI* repressor plasmid pAC017_cI (gray) and the transposon mutagenesis suicide plasmid pFG051 (yellow).

delivering the transposon, it also required the preparation of a higher volume of donor culture which could become limiting when performing very high-throughput experiments. Thus the 5:1 ratio was selected as it is more practical while keeping the efficiency very high (Fig 3D).

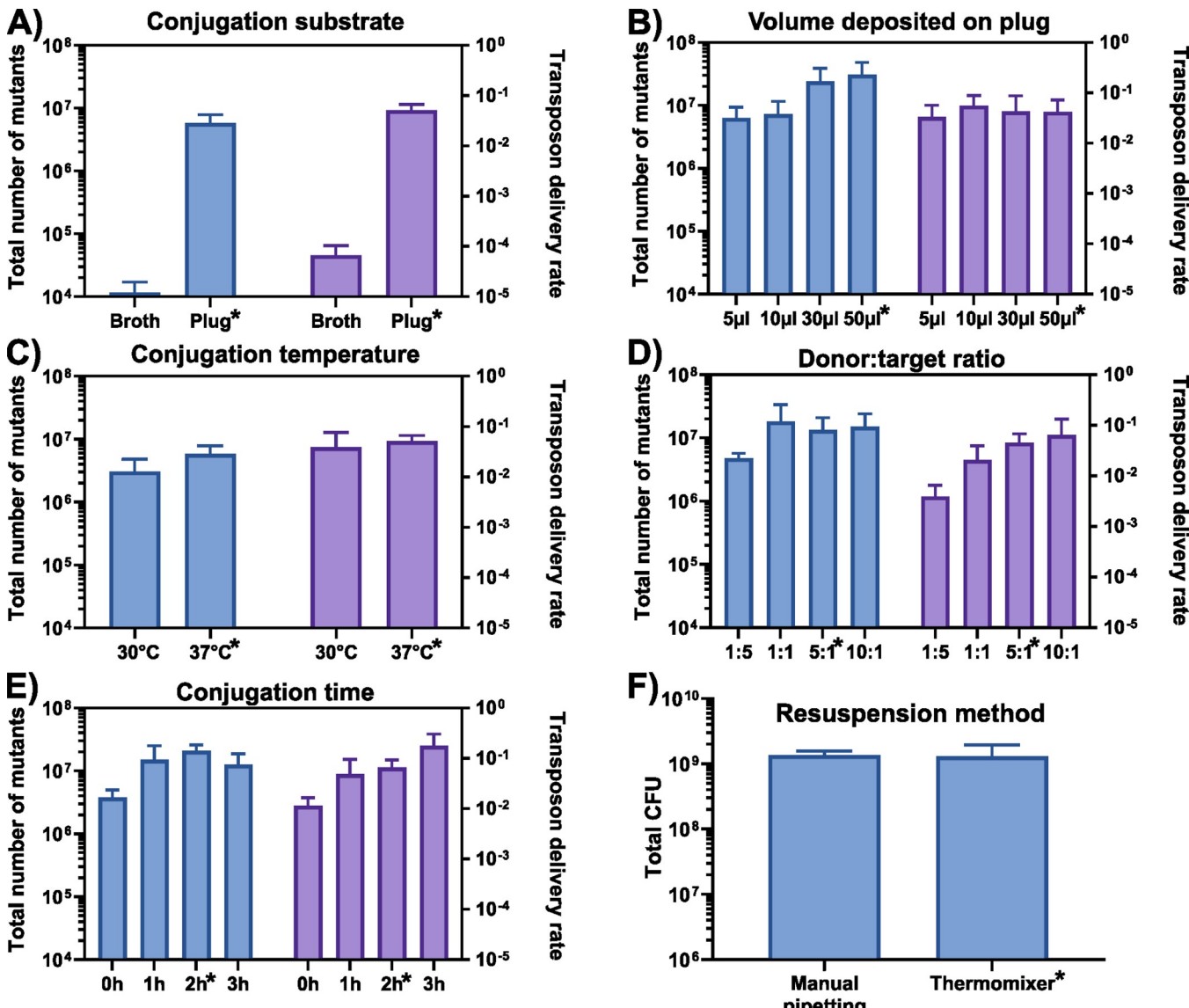

**Fig 3. Optimization of key HTTM parameters.** The total number of insertion mutants generated (blue) and the overall method efficiency (purple) were quantified while testing selected parameters such as **A)** the conjugation substrate (broth or agar plug), **B)** volume of conjugation mixture deposited on plug, **C)** conjugation temperature, **D)** the eAC494 donor to target cell ratio, and **E)** the duration of conjugative mating. **F)** Total number of colony forming units (CFU) recovered after the resuspension of the conjugative mixture using manual pipetting or the thermomixer procedure. The conditions selected for the standard HTTM protocol are indicated with an asterisk. Box and whiskers show mean and standard deviation across biological triplicates.

The conjugation time was also investigated and a 2h period was observed to be a good compromise between method efficiency and required time, thus allowing to perform the mutagenesis step in a regular workday (Fig 3E). Once the conjugation has been performed, the cells can be easily resuspended using a thermomixer with the same efficiency as the more tedious manual up and down pipetting procedure (Fig 3F). The mutants are next subjected to several passages over a few days to select those that can still grow and survive under the desired selective conditions despite the transposon insertions. To avoid bottlenecks during our selective passages we optimized the volume of culture transferred each day aiming to preserve at least 3 million mutants at each passage. This estimation is based on the cell concentrations typically obtained with the targeted bacterial strain grown in the liquid medium used for mutant selection. The exact volume of culture transferred at each passage should be optimized according to the specific target or growth conditions. The original pre-selection library can optionally be saved for future use by simply pooling together and making a glycerol stock of the usually discarded passage 1.

## DNA extraction

Following transposon mutagenesis, the genomic DNA (gDNA) from the generated pool of mutants can be harvested after each passage to monitor insertion levels as a function of time or simply at a defined time point to evaluate essentiality after a specific number of generations. DNA extraction is performed using a commercially available 96-well array of silica columns. To reduce the cost and ecological footprint associated with this step, the columns are regenerated after each extraction using a previously described method [18]. Briefly, the columns are first washed using an alkaline & Triton X-100 solution, the remaining DNA is degraded with a concentrated acid & Triton X-100 solution before a final extensive wash of the columns using the same alkaline solution as well as water to remove any trace of residual DNA. The amount of DNA recovered was consistent over successive cycles of extraction and column regeneration while no carry-over DNA was detected (S1, S2 Figs in S6 File). The recipes required for both DNA extraction and column regeneration solutions can be found in the online version of the protocol and the homemade versions were found to perform similarly to commercially available solutions (S3 Fig in S6 File). Using the current protocol, the DNA extraction yield for all wells was found to consistently range from 1.5 to 2.5 μg. This makes the systematic quantification unnecessary before sequencing library preparation as these amounts are well above the minimal suggested input of 30 ng of genomic DNA, which corresponds to approximately 6 million transposon insertion mutant genomes.

## Sequencing library preparation

Several sequencing platforms are available but given its widespread availability, the Illumina technology was selected for HTTM sequencing. The pFG051 plasmid was thus designed to contain an Illumina Nextera adapter sequence at one extremity of the transposon (Fig 2A), thereby positioning the start of sequencing reads exactly at the insertion site. This prevents any issue with cluster segregation due to the presence of identical sequences at the beginning of Illumina reads when reading through the transposon end and avoids dark cycles or the need to increase sequencing read diversity by spiking PhiX control DNA [19]. An additional advantage of this approach is the possibility of using shorter read lengths, which broadens options for kits and instruments that can be used.

The NEBNext Ultra II DNA Library Prep Kit for Illumina (NEB) was directly used for the gDNA fragmentation and ligation of adapters. However, the protocol was optimized to minimize reaction volumes, and purification was removed between most steps to reduce the time

and cost related to library preparation. These changes did not affect the yield and specificity of the transposon amplification (S4 Fig in S6 File) as long as the first PCR step contains a maximum of 2 μl of the unpurified ligation reaction in a 50 μl reaction (S5 Fig in S6 File). Since 8 μl of adapter-ligated DNA is available after the ligation step, up to four PCR replicates can be made to maximize the number of insertion sites. The HTTM samples are then pooled and sequenced using an Illumina sequencing system. The resulting data is then subjected to conventional bioinformatics quality control steps that include the adapter and quality trimming, alignment, etc. Duplicated genes or sequences (e.g., rRNA genes or identical insertion elements) are generally excluded from the analysis since sequencing reads cannot be unambiguously mapped. Bio-Tradis [20] or other toolkits [21, 22] then model the insertion densities across the genome into two distinct statistical populations, allowing the segregation of features likely essential from those that appear dispensable, respectively presenting low and high insertion densities (S6 Fig in S6 File).

## Material and methods

The protocols described in this peer-reviewed article is published on protocols.io:

- **Mutagenesis:** DOI: dx.doi.org/10.17504/protocols.io.36wgq72n3vk5/v1

- **gDNA extraction:** DOI: dx.doi.org/10.17504/protocols.io.q26g7yzz3gwz/v1

- **Library preparation:** DOI: dx.doi.org/10.17504/protocols.io.n2bvj8oowgk5/v1

They are included for printing as Supporting Information S1 File with this article. The exhaustive list of reagents and consumables used in this protocol is described in S2 File.

To facilitate high-throughput supernatant removal from deep-well plates, a custom adaptor for vacuum pump, the Aspir-8, has been designed and 3D printed. Models and printing instructions can be found at: https://www.thingiverse.com/thing:5569608. Aspir-8 allows rapid removal of all liquid in a 8-well row while preventing cross-contamination by using easily swappable p200 sterile tips. To further increase robustness and throughput, this adapter can be coupled with a guide that prevents pellet aspiration and leaves 50 μl of liquid at the bottom of the wells.

## Expected results

The HTTM method was optimized and validated using a GFP-expressing *E. coli* BW25113 [23] derivative by performing 24 biological replicates. The conjugative delivery of the transposon on agar plugs consistently generates >15 million mutants (S3 File) that can next be grown over many passages under selective conditions. The gDNA from the mutant pools can then be extracted at the desired passage and the sequencing libraries prepared in 96-well plates. For *E. coli* grown in EZ Rich medium [24], 5 consecutive passages (~18 divisions) were found to provide well-defined transposon insertion profiles (S7 Fig in S6 File) with an average of 261,309 ± 47,828 insertion sites per replicate at passage 5, corresponding to one insertion per ~16 bp (Fig 4A). These numbers are obtained by combining *in silico* the four PCR replicates obtained for each sample after the second PCR step of the Illumina library preparation protocol. Each supplementary PCR replicate adds a significant number of new sites but with diminishing returns as the number of replicates increases. On average, the number of different insertions raises by 61% by adding a second replicate, by 28% from a duplicate to a triplicate, and by 18% from a triplicate to a quadruplicate (Fig 4B & S8 Fig in S6 File). Interestingly, combining the four PCR replicates *in silico* and analyzing exactly 2 million reads per biological replicate would still provide 200,133 sites on average, representing 76% of the total insertions

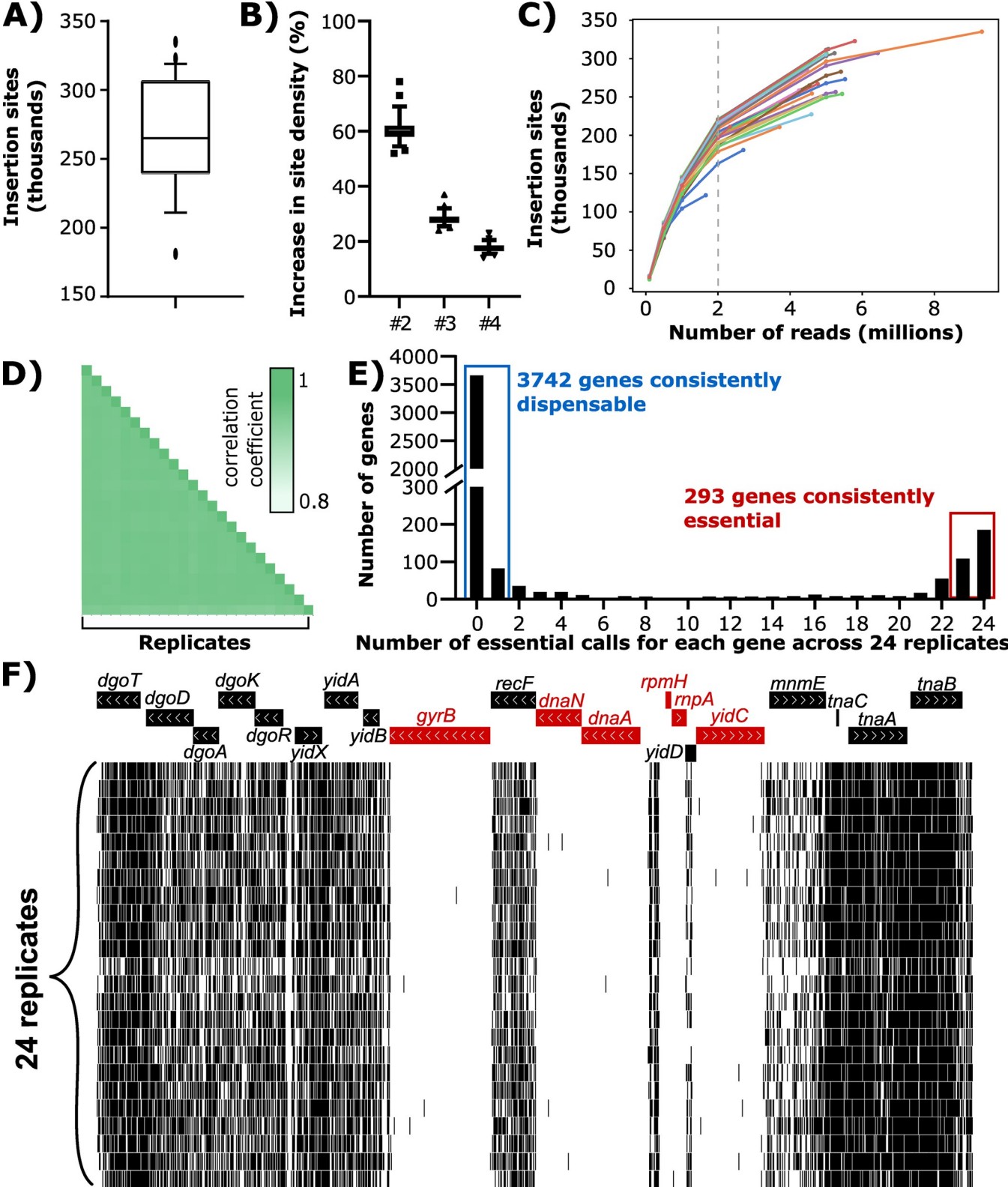

**Fig 4. Validation of the HTTM protocol in *E. coli* BW25113 using 24 biological replicates. A)** Number of transposon insertion sites detected in each replicate. Whiskers indicate 10–90 percentile. **B)** The average number of transposon insertion sites increases per additional PCR replicate included in the analysis. Whiskers indicate 10–90 percentile. **C)** Saturation curves showing the number of insertion sites as a function of the downsampling of each replicate.

**D)** Spearman correlation coefficients of the gene insertion densities between the 24 replicates. **E)** Frequency of essential gene status attribution for each gene across the 24 replicates. **F)** Example of known essential genes (in red) at a representative locus showing the insertion sites (black lines) in each of the 24 replicates.

(Fig 4C). This suggests that a relatively low sequencing depth is usable under these HTTM conditions. The rate of non-unique transposon insertions in a single bacterium using high-efficiency delivery by bacterial conjugatation was previously estimated to be approximately 15% (25) but few studies report this information, making comparison difficult between the different methods.

The reproducibility of the HTTM approach was next investigated by comparing the insertion density for each gene between the 24 replicates, showing a strong average Spearman correlation coefficient of 0.9425 ± 0.009 (Fig 4D & S4 File). The correlation coefficient stays almost identical when analyzing a subset of 2 million reads per biological sample (0.932 ± 0.007). Following gene essentiality calls in the *E. coli* BW25113 genome using the Bio-Tradis Toolkit [20], the different replicates were compared and 94% of all mappable genes have the same essentiality status across ≥23 samples (3742 consistently dispensable and 293 essential over the 4294 genes considered) (Fig 4E & S5 File). Interestingly, the remaining 6% have low insertion densities that are challenging to interpret for the Bio-Tradis toolkit and appear to be mostly composed of 1) genes that are very short (e.g., *eyeA* or *ssb*; see S9 Fig in S6 File) in which a single insertion could readily affect the essentiality status, or 2) gene interruptions leading to impaired fitness mutants that are not yet outcompeted by the rest of the population at passage 5 (e.g., the *sapABCDF* operon; see S10 Fig in S6 File). The data can also be visualized on a genome browser to inspect the sequencing read distribution and investigate the essentiality of specific features (Fig 4F).

The high number of samples that can be processed in parallel by HTTM allows large-scale comparison of gene essentiality in different growth conditions or organisms [25]. The higher throughput facilitates the incorporation of several replicates in gene essentiality studies. In our experience, up to 960 samples can be processed by a single person every week with high-reproducibility and at a low cost (<3$ per sample). Since the RP4 conjugative machinery has a broad host range [26–28] only minor adjustments to the pFG051 plasmid and protocol fine-tuning would be necessary to apply this approach to other microorganisms. By allowing the investigation of gene essentiality under a wide diversity of conditions, we expect that HTTM will contribute to a better understanding of microbial cell functioning, propose a role for many unknown function genes, and facilitate the design of highly-engineered bacterial genomes.

## Supporting information

**S1 File.**
(PDF)

**S2 File.**
(XLSX)

**S3 File.**
(XLSX)

**S4 File.**
(XLSX)

**S5 File.**
(XLSX)

**S6 File.**
(PDF)

## Acknowledgments

The authors are grateful to Dominick Matteau for his extensive review of the manuscript and Bruno Lemieux at the « Plateforme de purification des protéines » of the Université de Sherbrooke for the purification of the polymerase and the preparation of the Supermix 2X qPCR mix.

## Associated content

**Protocols on protocol.io DOI:**
- Mutagenesis: DOI: dx.doi.org/10.17504/protocols.io.36wgq72n3vk5/v1
- gDNA extraction: DOI: dx.doi.org/10.17504/protocols.io.q26g7yzz3gwz/v1
- Library preparation: DOI: dx.doi.org/10.17504/protocols.io.n2bvj8oowgk5/v1

**Aspir-8 3D model link**:
https://www.thingiverse.com/thing:5569608

## Author Contributions

**Conceptualization:** Antoine Champie, Amélie De Grandmaison, Frédéric Grenier, Pierre-Étienne Jacques, Sébastien Rodrigue.

**Data curation:** Antoine Champie, Amélie De Grandmaison, Simon Jeanneau, Frédéric Grenier.

**Formal analysis:** Antoine Champie, Amélie De Grandmaison, Simon Jeanneau, Frédéric Grenier.

**Funding acquisition:** Pierre-Étienne Jacques, Sébastien Rodrigue.

**Investigation:** Antoine Champie, Amélie De Grandmaison.

**Methodology:** Antoine Champie, Amélie De Grandmaison, Simon Jeanneau, Frédéric Grenier.

**Project administration:** Antoine Champie, Amélie De Grandmaison, Sébastien Rodrigue.

**Resources:** Antoine Champie, Amélie De Grandmaison, Pierre-Étienne Jacques.

**Software:** Simon Jeanneau, Frédéric Grenier, Pierre-Étienne Jacques.

**Supervision:** Antoine Champie, Sébastien Rodrigue.

**Validation:** Antoine Champie, Amélie De Grandmaison, Frédéric Grenier, Pierre-Étienne Jacques.

**Visualization:** Antoine Champie, Amélie De Grandmaison.

**Writing – original draft:** Antoine Champie, Amélie De Grandmaison.

**Writing – review & editing:** Antoine Champie, Amélie De Grandmaison, Pierre-Étienne Jacques, Sébastien Rodrigue.

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
