## [Decision Letter · Decision Letter 0]

19 Dec 2022

PONE-D-22-31056Enabling low-cost and robust essentiality studies with high-throughput transposon mutagenesis (HTTM)PLOS ONE

Dear Dr. Rodrigue,

Thank you for submitting your manuscript to PLOS ONE. After careful consideration, we feel that it has merit but does not fully meet PLOS ONE’s publication criteria as it currently stands. Therefore, we invite you to submit a revised version of the manuscript that addresses the points raised during the review process.

We look forward to receiving your revised manuscript.

Kind regards,

Cinzia Calvio, PhD

Academic Editor

PLOS ONE

Journal Requirements:

Reviewers' comments:

Reviewer's Responses to Questions

**Comments to the Author**

1. Does the manuscript report a protocol which is of utility to the research community and adds value to the published literature?

Reviewer #1: Yes

Reviewer #2: Yes

2. Has the protocol been described in sufficient detail?

To answer this question, please click the link to protocols.io in the Materials and Methods section of the manuscript (if a link has been provided) or consult the step-by-step protocol in the Supporting Information files.

The step-by-step protocol should contain sufficient detail for another researcher to be able to reproduce all experiments and analyses.

Reviewer #1: Yes

Reviewer #2: Yes

3. Does the protocol describe a validated method?

Reviewer #1: Yes

Reviewer #2: Yes

4. If the manuscript contains new data, have the authors made this data fully available?

Reviewer #1: Yes

Reviewer #2: No

**5. Is the article presented in an intelligible fashion and written in standard English?**

Reviewer #1: Yes

Reviewer #2: Yes

6. Review Comments to the Author

Reviewer #1: In this manuscript, the authors report an interesting method High-Throughput Transposon Mutagenesis (HTTM) protocol which is robust and inexpensive. HTTM reliably provides provides high insertion densities with impressive reproducibility. This article is important to the researchers in Transposon Insertion Sequencing field and will help progress the field forward. There are, however, several issues in the review that require the author's attention.

1、 Introduction.

“The combination of transposon mutagenesis with next-generation sequencing” should be described as “Transposon-insertion sequencing (TIS) methods” not “Tn-seq”.

Tn-seq based on the assembly of a saturated Mariner transposon (not Tn5) insertion library was presented by Tim van Opijnen and Henry L. Levin. Transposon sequencing (Tn-Seq), transposon-directed insertion site sequencing (TraDIS), insertion sequencing (INSeq) and high-throughput insertion tracking by deep sequencing (HITS) are four variations on the TIS method. Authors may refer to recent reviews by Amy K. Cain et al. and Tim van Opijnen et al., doi: 10.1038/s41576-020-0244-x, doi: 10.1146/annurev-genet-112618-043838.

2、 A transposon mutagenesis plasmid contains an oriTRP4 sequence enabling its conjugative transfer into bacterial cells targeted for mutagenesis to generate the huge number of mutants. The question is whether we can confirm that each bacterium carries only one single transposon insertion. We know, if a bacterium contains multiple gene mutations it may misestimate the fitness of gene after library selection. In Tn-seq technology, in vitro transposition and subsequent transformation are both low-frequency events, so it is believed that the overwhelming majority of transformants harbor single transposon insertions.

3、HTTM depends on the frequency of conjugative transfer of RP4 plasmid between bacteria, so can it be easy applied to other microbial species?

Reviewer #2: General comments:

Transposon mutagenesis coupled with next-generation sequencing (Tn-seq) has been a powerful high-throughput genetic screening method for functional genomics in bacteria. However, the generation of mutant libraries can be labour intensive and expensive. In this manuscript, the authors offer a complete protocol from making mutant library, DNA extraction to Illumina sequencing which is claimed to reduce cost and labour. While I have no major issues with the protocol and its feasibility, I do have several minor points that could benefit from further clarification.

1. Estimation of independent mutants: the authors stated that from each agar plug, they could obtain ~15 million mutants. I assume that this number was simple obtained by CFU count and a fair number of these CFU were siblings, leading to the “inflated” number of ~15 million mutants, yet when sequenced only showed ~261,309 insertion sites. If we assume that one insertion site represents one independent mutant, then the initial estimation of 15 million mutants is >50x over. Indeed, based on the statement in lines 230-231, I would interpret that I will need to pool 3-4 replicates to obtain a well-saturated mutant library. This should be clarified in the manuscript.

2. The use of Pearson correlation coefficient is sensitive to sample size and sample distribution. It is well-known (also shown in Fig S6) that insertion index/insertion density per gene is bimodal distributed while Pearson correlation works best for Gaussian distribution. I suggest that the authors use a more appropriate measurement for showing correlation. A supplementary figure of Bland-Altman plots might do.

3. The bias of mutant libraries: the mutagenesis protocol includes passaging for several days to select for the transposon mutants and remove the donors. However, these passages also adapt the transposon mutant population to this particular growth condition. The resulting mutant libraries are therefore not good starting libraries to be used to investigate a different challenge/growth condition. One might need to make new mutant libraries for each condition examined. This point should be discussed in the manuscript.

4. The link to raw data https://www.ncbi.nlm.nih.gov/sra/PRJNA896618 does not work (Error message: The following term was not found in SRA: PRJNA896618)

7. PLOS authors have the option to publish the peer review history of their article (what does this mean?). If published, this will include your full peer review and any attached files.

Reviewer #1: No

Reviewer #2: No

---

## [Author Response · Author response to Decision Letter 0]

6 Feb 2023

--We thank the reviewers for their precious comments and insights We appreciate the constructive remarks that strengthen our manuscript. The original comments are presented below in black. Our responses are shown in blue. 

Reviewer #1: General comments:

In this manuscript, the authors report an interesting method High-Throughput Transposon Mutagenesis (HTTM) protocol which is robust and inexpensive. HTTM reliably provides high insertion densities with impressive reproducibility. This article is important to the researchers in Transposon Insertion Sequencing field and will help progress the field forward. There are, however, several issues in the review that require the author's attention.

1 Introduction.“The combination of transposon mutagenesis with next-generation sequencing” should be described as “Transposon-insertion sequencing (TIS) methods” not “Tn-seq”.Tn-seq based on the assembly of a saturated Mariner transposon (not Tn5) insertion library was presented by Tim van Opijnen and Henry L. Levin. Transposon sequencing (Tn-Seq), transposon-directed insertion site sequencing (TraDIS), insertion sequencing (INSeq) and high-throughput insertion tracking by deep sequencing (HITS) are four variations on the TIS method. Authors may refer to recent reviews by Amy K. Cain et al. and Tim van Opijnen et al., doi:10.1038/s41576-020-0244-x, doi: 10.1146/annurev-genet-112618-043838.

--The transposon mutagenesis nomenclature has been updated in the abstract and the introduction to use TIS instead of Tn-seq. The suggested review articles have also been cited. (see lines 49 and 69 of the track-change mode document)

2 A transposon mutagenesis plasmid contains an oriTRP4 sequence enabling its conjugative transfer into bacterial cells targeted for mutagenesis to generate the huge number of mutants. The question is whether we can confirm that each bacterium carries only one single transposon insertion. We know, if a bacterium contains multiple gene mutations it may misestimate the fitness of gene after library selection. In Tn-seq technology, in vitro transposition and subsequent transformation are both low-frequency events, so it is believed that the overwhelming majority of transformants harbor single transposon insertions.

--We expect our approach based on conjugation will likely have higher double transposition occurrence than other approaches that typically generate a lower number of mutants, which lower the chance of non-unique transposon insertion events. We previously tested a similar system based on the conjugation using the oriTRP4 and found that roughly 85% of all mutants (3/20 colonies tested) harboured a single transposon insertion (https://savoirs.usherbrooke.ca/handle/11143/11637). We now cite this work at lines 270-273 of the revised manuscript to indicate the possibility that a cell contains more than one transposon. Recent studies in E. coli, for example https://doi.org/10.1128/mBio.02096-17, using in vitro transposition do not disclose the rates of non-unique transposition events in a cell, making direct comparison difficult. 

3 HTTM depends on the frequency of conjugative transfer of RP4 plasmid between bacteria, so can it be easy applied to other microbial species?

--The RP4 plasmid can transfer at relatively high rates in many organisms such as E. coli, Salmonella, Klebsiella, Pseudomonas, Vibrio, Acinetobacter, Xanthomonas, Rhizobium, Sphingomonas etc. For example, Gram-negative organisms like Acinetobacter and Sphingomonas (https://doi.org/10.1111/j.1574-6968.1999.tb13543.x), the RP4 conjugation system already displays a high transferfrequency (>1x10-2) suggesting that the HTTM system is already applicable. For Gram-positive organisms like Lactobaccilus casei, transfer rates (1.4x10-2) have already been observed () while in other like Staphylococcus aureus or eukaryotes much lower efficiencies have been reported: 2x10-8 to 5x10-5 (https://doi.org/10.1111/j.1574-6968.1987.tb02558.x, https://doi.org/10.1128%2Fjb.180.24.6538-6543.1998) suggesting that another conjugative system may be required to carry the transposon or that the RP4 conjugative system requires an accelerated evolution (https://doi.org/10.15252/msb.202110335) to perform efficiently towards those recipients. Overall, the system is likely to be already usable without any modification in a many organisms but will need to be adapted to perform adequately in others. 

This point has been clarified in the conclusion (Lines 297-298)

Reviewer #2: General comments:

Transposon mutagenesis coupled with next-generation sequencing (Tn-seq) has been a powerful high-throughput genetic screening method for functional genomics in bacteria. However, the generation of mutant libraries can be labour intensive and expensive. In this manuscript, the authors offer a complete protocol from making mutant library, DNA extraction to Illumina sequencing which is claimed to reduce cost and labour. While I have no major issues with the protocol and its feasibility, I do have several minor points that could benefit from further clarification.

1. Estimation of independent mutants: the authors stated that from each agar plug, they could obtain ~15 million mutants. I assume that this number was simple obtained by CFU count and a fair number of these CFU were siblings, leading to the “inflated” number of ~15 million mutants, yet when sequenced only showed ~261,309 insertion sites. If we assume that one insertion site represents one independent mutant, then the initial estimation of 15 million mutants is >50x over. Indeed, based on the statement in lines 230-231, I would interpret that I will need to pool 3-4 replicates to obtain a well-saturated mutant library. This should be clarified in the manuscript.

--The number of transposon insertion mutants is calculated by estimating colony forming units (CFU) immediately after the 2h conjugation between the eAC494 donor strain and the target bacteria. A fraction of the ~15 million mutants are certainly identical for at least two reasons: 

1) The insertion of transposon can occur at the same site in two different cells, which would not be so surprising given that every transposon has a slight bias as discussed in the manuscript (see lines 77-83);

2) Some of the mutants may divide during the 2h conjugation period although these conditions are presumably not optimal for cell doubling;

However, the average of 261,309 insertions sites mentioned at lines 230-231 were obtained after 5 culture passages and do not represent the initial number of mutants. Only a subset of the initial mutants (approximately 9 million) obtained during conjugation are introduced in the initial passage and many mutants will be outcompeted and disappear from the population after a few generations or passages. It is therefore expected that the number of insertion sites observed at each passage will decrease through time.

The diversity of the initial library (immediately after conjugation) has not been estimated by sequencing, and the closest sequencing-based estimate we can provide is the number of different mutants present after 1 selective passage with approximately 440K mutants at a sequencing depth of 2 million reads. 

The replicates mentioned on line 230-231 are technical replicates of a part of the library preparation protocol and are intended to extract more diversity from the same, already existing, pool of mutants while limiting potential PCR bias and are not related to the number of mutants present in the initial pool. This has been clarified in the at lines 24-265 of the revised manuscript (in track change mode). 

2. The use of Pearson correlation coefficient is sensitive to sample size and sample distribution. It is well-known (also shown in Fig S6) that insertion index/insertion density per gene is bimodal distributed while Pearson correlation works best for Gaussian distribution. I suggest that the authors use a more appropriate measurement for showing correlation. A supplementary figure of Bland-Altman plots might do.

--We thank the reviewer for this comment. We now present the average Spearman correlation coefficient for the 24 replicates in the text along with the complete data in the supplementary material (see Lines 57, 275, 277, and 288-289). 

3. The bias of mutant libraries: the mutagenesis protocol includes passaging for several days to select for the transposon mutants and remove the donors. However, these passages also adapt the transposon mutant population to this particular growth condition. The resulting mutant libraries are therefore not good starting libraries to be used to investigate a different challenge/growth condition. One might need to make new mutant libraries for each condition examined. This point should be discussed in the manuscript.

--The HTTM protocol as described in our manuscript is designed to have any growth condition investigated in parallel and avoids the step of stocking a library for future use. The stock performed after passage 4 is merely a safeguard in case of problem during the DNA extraction and is not intended to be used for future experiments. The generation of a reusable mutant library used to be part of an earlier version of the protocol and can be performed by simply pooling and cryopreserving the transposon mutagenesis leftover from each well instead of discarding it after the initial library growth. We revised the text to mention this possibility at line 193-194.

4. The link to raw data https://www.ncbi.nlm.nih.gov/sra/PRJNA896618 does not work (Error message: The following term was not found in SRA: PRJNA896618)

--SRA has been contacted and the raw data is now released.

---

## [Decision Letter · Decision Letter 1]

21 Mar 2023

Enabling low-cost and robust essentiality studies with high-throughput transposon mutagenesis (HTTM)

PONE-D-22-31056R1

Dear Dr. Rodrigue,

We’re pleased to inform you that your manuscript has been judged scientifically suitable for publication and will be formally accepted for publication once it meets all outstanding technical requirements.

Kind regards,

Cinzia Calvio, PhD

Academic Editor

PLOS ONE

Additional Editor Comments (optional):

Reviewers' comments:

Reviewer's Responses to Questions

**Comments to the Author**

1. Does the manuscript report a protocol which is of utility to the research community and adds value to the published literature?

Reviewer #2: Yes

2. Has the protocol been described in sufficient detail?

To answer this question, please click the link to protocols.io in the Materials and Methods section of the manuscript (if a link has been provided) or consult the step-by-step protocol in the Supporting Information files.

The step-by-step protocol should contain sufficient detail for another researcher to be able to reproduce all experiments and analyses.

Reviewer #2: Yes

3. Does the protocol describe a validated method?

Reviewer #2: Yes

4. If the manuscript contains new data, have the authors made this data fully available?

Reviewer #2: Yes

**5. Is the article presented in an intelligible fashion and written in standard English?**

Reviewer #2: Yes

6. Review Comments to the Author

Reviewer #2: I thank the authors for incorporating my suggestions into their revised manuscript. I don't have any further comments.

7. PLOS authors have the option to publish the peer review history of their article (what does this mean?). If published, this will include your full peer review and any attached files.

Reviewer #2: No

---

## [Editor Report · Acceptance letter]

31 Mar 2023

PONE-D-22-31056R1 

Enabling low-cost and robust essentiality studies with high-throughput transposon mutagenesis (HTTM) 

Dear Dr. Rodrigue:

I'm pleased to inform you that your manuscript has been deemed suitable for publication in PLOS ONE. Congratulations! Your manuscript is now with our production department. 

Kind regards, 

on behalf of

Dr Cinzia Calvio 

Academic Editor

PLOS ONE